# Small Intestinal Bacterial Overgrowth Is Associated with Poor Prognosis in Cirrhosis

**DOI:** 10.3390/microorganisms11041017

**Published:** 2023-04-13

**Authors:** Irina Efremova, Roman Maslennikov, Aliya Alieva, Elena Poluektova, Vladimir Ivashkin

**Affiliations:** 1Department of Internal Medicine, Gastroenterology and Hepatology, Sechenov University, 119121 Moscow, Russia; 2The Interregional Public Organization “Scientific Community for the Promotion of the Clinical Study of the Human Microbiome”, 119121 Moscow, Russia

**Keywords:** SIBO, gut microbiota, bacterial translocation, prognosis, gut microbiome

## Abstract

Background: Small intestinal bacterial overgrowth (SIBO) is associated with numerous manifestations of cirrhosis. To determine whether the presence of SIBO affects the prognosis in cirrhosis was the aim of the study. Methods: This prospective cohort study included 50 patients. All participants underwent a lactulose hydrogen breath test for SIBO. The follow-up period was 4 years. Results: SIBO was detected in 26 (52.0%) patients: in 10 (52.6%) patients with compensated cirrhosis and in 16 (51.6%) ones with decompensated cirrhosis. Twelve (46.2%) patients with SIBO and four (16.7%) patients without SIBO died within 4 years (*p* = 0.009). Among patients with decompensated cirrhosis, 8 (50.0%) patients with SIBO and 3 (20.0%) patients without SIBO died (*p* = 0.027). Among patients with compensated cirrhosis, four (40.0%) patients with SIBO and one (11.1%) patient without SIBO died (*p* = 0.045). Among patients with SIBO, there was no difference in mortality between patients with compensated and decompensated cirrhosis (*p* = 0.209). It was the same for patients without SIBO (*p* = 0.215). SIBO affects the prognosis only in the first year of follow-up in decompensated cirrhosis, and only in subsequent years in compensated cirrhosis. Presence of SIBO (*p* = 0.028; HR = 4.2(1.2–14.9)) and serum albumin level (*p* = 0.027) were significant independent risk factors for death in cirrhosis. Conclusions: SIBO is associated with poor prognosis in cirrhosis.

## 1. Introduction

The gut–liver axis plays an important role in the pathogenesis of cirrhosis [1]. Although maximum attention is focused on the study of changes in the composition of the gut microbiome (gut dysbiosis) now, its determination has not become a part of clinical practice due to the complexity and high cost of this test [2]. However, the determination of small intestinal bacterial overgrowth (SIBO), another type of gut microbiota pathology, is available in many clinics around the world. SIBO is detected in almost half of patients with cirrhosis [3].

It is believed that changes in the gut microbiota (gut dysbiosis) and its expansion into the small intestine (SIBO) lead to bacterial translocation (the transfer of bacteria and their components from the intestinal contents to the macroorganism), systemic inflammation and hemodynamic changes that contribute to the development of complications of cirrhosis: ascites, hepatic encephalopathy, esophageal varices, spontaneous bacterial peritonitis and others [4]. This predetermines a poor prognosis in patients with SIBO. It has already been shown that SIBO is associated with the development of ascites, minimal hepatic encephalopathy, spontaneous bacterial peritonitis, bacterial translocation, and hemodynamic changes in cirrhosis [3,5]. However, no study has yet been published that has examined the effect of SIBO presence on the long-term prognosis in cirrhosis. This became the aim of our work.

## 2. Materials and Methods

### 2.1. Participants

We screened to include patients with cirrhosis admitted to the clinic out from March 2016 to December 2016. The study was approved by the institutional ethics board (protocol NO. 03-16), and conducted in accordance with the Declaration of Helsinki. Informed consent was obtained from all individual participants.

We considered for inclusion inpatients at least 18 years of age, who were diagnosed with cirrhosis based on clinical, biochemical, and ultrasound findings, and further verified by histology. The patients who had used lactulose, lactitol, or other prebiotics, probiotics, antibiotics, or prokinetics or consumed alcohol in the past 6 weeks, and had current infection or diabetes, inflammatory bowel disease, cardiac disease, cancer, or any other disease considered to be severe were excluded.

There was no data that could be used to calculate the required sample size.

### 2.2. Diagnostic Workup

The severity of liver disease was determined using the Child–Turcotte–Pugh (CTP) scoring system in which Class A is defined as compensated cirrhosis, and Classes B and C are defined as decompensated cirrhosis [6].

The lactulose hydrogen breath test was used for SIBO diagnosis as the North American Consensus and the national scientific organization recommended [7,8].

We used a Gastrolyzer (Bedfont, The United Kingdom) to measure the breath samples. The patient consumed 10 g lactulose dissolved in 200 mL of water, after which the hydrogen content in the exhaled air was determined every 15 min for 90 min. Just prior to the consumption of lactulose, the baseline level of hydrogen in the exhaled air was also measured. We considered the presence of SIBO when there was an increase in breath hydrogen of at least 20 ppm above the baseline value within 90 min. 

Patients were divided into groups according to the presence or absence of SIBO (C-SIBO(+) and C-SIBO(−), respectively) and further into subgroups according to the degree of cirrhosis compensation.

### 2.3. Follow-Up

Each patient was contacted every 3 months to confirm that this patient was alive by phone. If there was no answer, we contacted the patient’s relatives by phone to find out if the patient was alive or dead. If it was not possible to contact them, we studied patient electronic medical record in the united medical information and analytical system, in which the registrations of death are entered. The follow-up period was 4 years. 

### 2.4. Outcomes

The primary outcome was all-cause mortality. The components of the Child–Turcotte–Pugh scale (total bilirubin, prothrombin and albumin levels, ascites and hepatic encephalopathy stage), esophageal varicose stage, and SIBO presence were selected as predictors. Secondary outcomes were the mortalities of patients in the first and subsequent years of follow-up.

We performed a subgroup analysis, separating patients with compensated and decompensated cirrhosis, with and without SIBO.

We tried to obtain all the data. If more than 25% of the data were missing, the analysis was not performed. If less than 25% of the data were missing, only the received data were analyzed.

### 2.5. Statistical Analysis

Statistical analysis was performed with STATISTICA 10 software (StatSoft Inc., Tulsa, OK, USA). The difference between continuous variables was assessed using the Mann–Whitney test. Data are presented as median [interquartile range]. Fisher’s exact test was used to assess the difference between categorical variables. Survival was assessed using the Kaplan–Meier estimator and Mantel–Cox test. A Cox regression model was used to assess the influence of factors on patient survival and hazard ratio (HR). A *p*-value < 0.05 was considered statistically significant.

## 3. Results

Among screened patients, 50 met the inclusion criteria and were enrolled in the study (Figure 1).

SIBO was detected in 26 (52%) patients that make up group C-SIBO(+). Patients without SIBO were included in the group C-SIBO(−) (Figure 1). Groups C-SIBO(+) and C-SIBO(−) were comparable in age, body mass index, sex distribution, severity of cirrhosis, and other characteristics (Table 1). There was no significant difference in the drugs used between these groups (Table 2).

Nineteen patients had compensated cirrhosis (CTP class A) and the remaining 31 had decompensated cirrhosis (19 CTP class B and 12 CTP class C). There was no significant difference in the etiology of cirrhosis between patients with compensated and decompensated cirrhosis. SIBO was detected in 10 (52.6%) of patients with compensated cirrhosis and in 16 (51.6%) of ones with decompensated cirrhosis. Patients with compensated cirrhosis and SIBO formed subgroup CC-SIBO(+), while patients with decompensated cirrhosis and SIBO formed subgroup DC-SIBO(+). Patients with compensated cirrhosis without SIBO formed subgroup CC-SIBO(−), while patients with decompensated cirrhosis without SIBO formed subgroup DC-SIBO(−) (Figure 1). 

There were no patients with missing data. All data on patient survival during the entire follow-up period were obtained.

Sixteen (32.0%) patients died within 4 years (Figure 2a). The causes of death in all patients were acute-on-chronic liver failure, the development of which in 75% of cases was preceded by bleeding from esophageal varices. Fatal bleeding from esophageal varices occurred in 9 (35.6%) patients with SIBO and in 3 (12.5%) patients without SIBO (*p* = 0.066).

The mortality rate of patients with decompensated cirrhosis during the first year of follow-up was higher than that of patients with compensated cirrhosis (22.6% vs. 0.0%; *p* = 0.024). However, there was no significant difference in mortality between patients with decompensated and compensated cirrhosis, if it was evaluated over the entire four-year follow-up period (35.5% vs. 26.3%; *p* = 0.178).

Mortality in patients with SIBO was higher both in the general group (46.2% vs. 16.7%; *p* = 0.009; Figure 2b) and in the groups with decompensated (50.0% vs. 20.0%; *p* = 0.027; Figure 2c) and compensated (40.0% vs. 11.1%; *p* = 0.045; Figure 2d) cirrhosis.

Among patients with decompensated cirrhosis, mortality was higher in patients with SIBO than in patients without SIBO during the first year of follow-up (37.5% vs. 6.7%; *p* = 0.010), but it did not differ significantly during subsequent years of follow-up (12.5% vs. 13.3%; *p* = 0.459).

Among patients with SIBO, there was no difference in mortality between patients with compensated and decompensated cirrhosis (40.0% vs. 50.0%; *p* = 0.209; Figure 2e). It was the same for patients without SIBO (11.1% vs. 20.0%; *p* = 0.215; Figure 2f). 

Among patients with SIBO, patients with compensated cirrhosis tended to die later than patients with decompensated cirrhosis (2.8 (1.8–3.4) vs. 0.6 (0.5–2.0) years; *p* = 0.102). Among the deceased patients with decompensated cirrhosis and SIBO, six (75.0%) died during the first year of follow-up. All deceased patients with compensated cirrhosis and SIBO died after 1 year of follow-up (*p* = 0.030).

Among patients with SIBO, mortality during the first year of follow-up was higher in the group of patients with decompensated cirrhosis (37.5% vs. 0.0%; *p* = 0.033), and during subsequent years in the group of patients with compensated cirrhosis (40.0% vs. 12.5%; *p* = 0.045; Figure 2e).

Three sections can be distinguished in the overall survival curve (Figure 2a): the first one is a sharp descent during the first year of follow-up, which accounts for 43.8% of the deaths, then a plateau during the second and third years of follow-up, which accounts for only 12.5% of the deaths, and a second sharp descent during the fourth year of follow-up, accounting for 43.8% of deaths. Moreover, 85.7% of those who died during the first descent of the survival curve were included in the DC-SIBO(+) group, 100% of those who died during the plateau were in the CD-SIBO(+) group, and those who died during the second descent of the curve were approximately evenly distributed between all four groups.

Multivariate regression analysis revealed that the presence of SIBO and albumin level in the blood were significant independent risk factors for death in our patients (Table 3).

## 4. Discussion

The gut–liver axis has a great importance in the progression of cirrhosis. The following main changes occur in the intestine in cirrhosis: the proportion of harmful endotoxin-containing bacteria in the gut microbiota increases and the proportion of beneficial bacteria decreases (gut dysbiosis [9,10,11,12,13,14,15,16]), the colon microbiota expands into the small intestine (SIBO) and the permeability of the intestinal barrier increases. All this contributes to bacterial translocation that is the penetration of bacteria and their components from intestinal content into macroorganisms. It causes the development of systemic inflammation, which leads to splanchnic vasodilation. The latter leads to a decrease in blood pressure, water and sodium retention, and the development of hyperdynamic circulation. All of this enhances portal hypertension, contributing to the development of ascites, spontaneous bacterial peritonitis, portal blood shunting, hepatic encephalapathy and esophageal varices, and other complications of cirrhosis that increase mortality [4,17].

Hyperdynamic circulation in cirrhosis has been described for a long time [18,19], but only recently its role in the pathogenesis of complications of cirrhosis has become clearer [20]. 

A recent study has shown that SIBO and gut dysbiosis are distinct forms of gut microbiota pathology and should be considered separately [21].

Among the three main pathogenetic links of the gut–liver axis described above (gut dysbiosis, SIBO and increased permeability), SIBO can only be determined in clinical practice. A number of guidelines for the diagnosis and treatment of SIBO in real clinical practice have already been published [8,22,23,24]. A meta-analysis [3] showed that SIBO is associated with ascites [25,26,27,28,29], hepatic encephalopathy [30,31,32], and spontaneous bacterial peritonitis [25,26,28,33,34,35,36] in cirrhosis. We also found that SIBO is associated with characteristic hemodynamic changes in decompensated cirrhosis. However, even in compensated cirrhosis, it is associated with dilatation of large arterial vessels, which may be the first stage of splanchnic vasodilation [5].

However, despite the presence of these associations, no studies have been published that described the long-term prognosis of patients with cirrhosis, depending on the presence of SIBO. Our study is the first to answer this question and thus sums up SIBO research in cirrhosis. This is its strength.

We have shown that the presence of SIBO is associated with a poor prognosis in both decompensated and compensated cirrhosis. Moreover, the long-term prognosis was determined more by SIBO than by the degree of compensation of liver function. Thus, there was no difference in mortality among patients with compensated and decompensated cirrhosis if they have SIBO. However, the analysis of the survival curve showed that patients with decompensated cirrhosis and SIBO, as a rule, died in the first year of follow-up, after which the survival curve reached a plateau. The opposite picture was observed in compensated cirrhosis: the survival rate of these patients was significantly lower than in decompensated cirrhosis in the first year, but their mortality significantly increased in subsequent years.

Such differences can be explained by the fact that in decompensated cirrhosis, the barrier function of the intestine significantly reduces, leading to massive bacterial translocation, which triggers the pathogenetic mechanism described above. This barrier function is still preserved in compensated cirrhosis, but its progressive decrease takes place, which leads to decompensation of cirrhosis in patients with SIBO after a few years and death.

The transition of the mortality curve of patients with decompensated cirrhosis to a plateau after 1 year is very interesting. The presence of SIBO determined mortality of these patients during the first year of follow-up, but then this ceased to have a significant impact on their prognosis. It is possible that a state of intestinal permeability or the absence of significant gut dysbiosis protected those patients with decompensated cirrhosis and SIBO, who survived the first year of follow-up, from the harmful effects of SIBO. Further research is needed to answer this question.

The question of the management of patients with cirrhosis and SIBO also remains open. The use of antibiotics in the treatment of SIBO leads only to its temporary elimination, and after a few months it returns again, since the cause of its development, which has not yet been precisely established, is not eliminated. It is assumed that this cause may be slowed intestinal motility [3], possibly due to portal hypertension. Further research is needed to clarify the cause for the development of SIBO in cirrhosis and to develop its optimal management.

The limitation of our study is the small number of its participants, which did not prevent us from obtaining significant results and making extracts based on them. The second limitation is that the hydrogen breath test criteria for SIBO diagnosis are poorly standardized. The third limitation of the study is that we were able to determine the cause of death in less than half of the cases, which did not allow us to conduct a correct analysis of them. The forth limitation is that we could not guarantee that during the follow-up period patients did not take drugs that eliminate SIBO. However, given that SIBO tends to recur, we hope that this limitation did not have a significant impact on our study results. It would be very interesting to compare the incidence of various complications of cirrhosis (spontaneous bacterial peritonitis, hepatic encephalopathy, and others) in our patients, but since this was not part of the initial tasks of the study, statistics on them are also incomplete. 

## 5. Conclusions

In conclusion, we have showed that SIBO is associated with a poor prognosis in cirrhosis. According to our preliminary data, SIBO affected the prognosis only in the first year of follow-up in decompensated cirrhosis, and only in subsequent years in compensated cirrhosis. Further studies are required to verify our results in a larger cohort of patients and to select the optimal drugs and regimes for their therapeutic and prophylactic use in order to improve the prognosis of cirrhosis patients with SIBO.

## Figures and Tables

**Figure 1 microorganisms-11-01017-f001:**
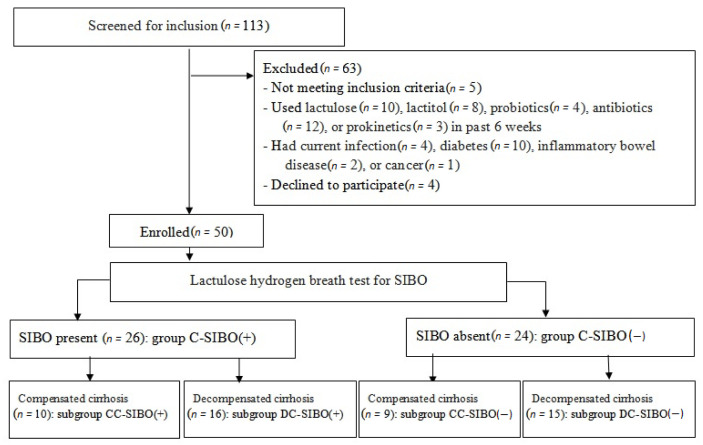
CONSORT 2010 flow diagram.

**Figure 2 microorganisms-11-01017-f002:**
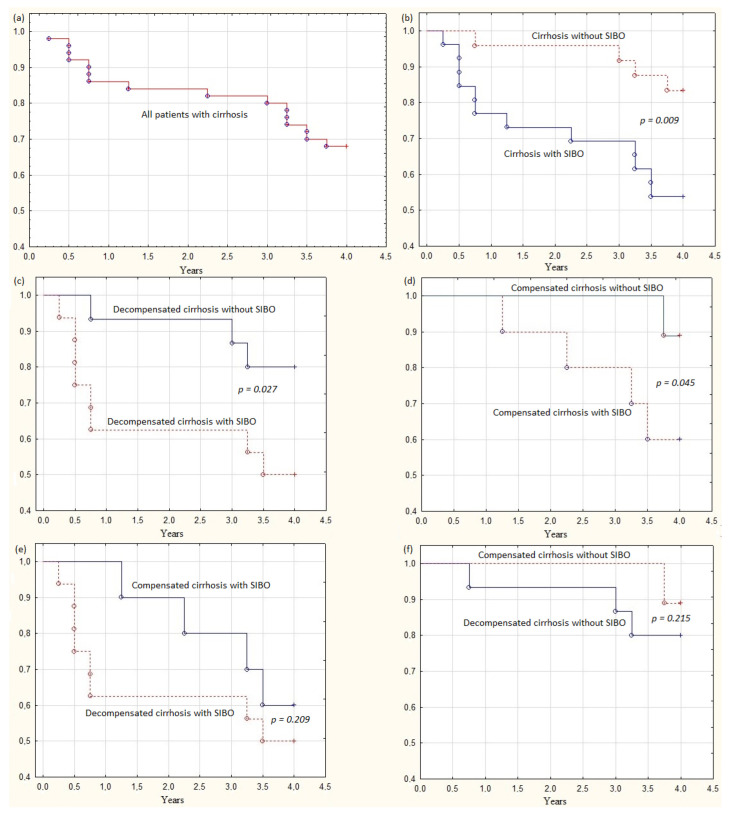
Survival curves of patients: (**a**) all patients; (**b**) cirrhosis with small intestinal bacterial overgrowth (SIBO) and without SIBO; (**c**) decompensated cirrhosis with and without SIBO; (**d**) compensated cirrhosis with and without SIBO; (**e**) patients with SIBO and with compensated and decompensated cirrhosis; (**f**) patients without SIBO with compensated and decompensated cirrhosis.

**Table 1 microorganisms-11-01017-t001:** Main characteristics of patients with small intestinal bacterial overgrowth (SIBO) and without it.

	Patients with SIBO (*n* = 26)	Patients without SIBO (*n* = 24)	*p* *
Age, years	49 [39–59]	52 [39–58]	0.904
Body mass index, kg/m^2^	24.6 [22.7–27.8]	24.1 [22.5–26.8]	0.636
Men/women	13/13	11/13	0.785
Race (Caucasian/other)	25/1	24/0	1.000
Etiology of cirrhosis: alcohol	9	9	1.000
autoimmune	2	9	0.016
viral	11	6	0.242
cryptogenic	4	0	0.111
Child–Turcotte–Pugh score	8 [6–10]	8 [6–9]	0.465
Model for End-Stage Liver Disease score	10.5 [7.4–13.5]	9.2 [7.1–11.9]	0.317
Esophageal varices (present/absent)	20/6	20/4	0.728
Hepatic encephalopathy (overt/minimal/ absent)	9/9/8	8/11/5	0.526
Ascites (present/absent)	18/8	10/14	0.086
Serum albumin, g/L	35 [30–40]	36 [31–41]	0.505
Serum total bilirubin, μmol/L	36.8 [27.9–55.2]	36.6 [23.2–64.0]	0.954
Prothrombin index (Quick test), %	57 [50–68]	65 [55–71]	0.146
Creatinine, mg/dL	0.74 [0.60–0.91]	0.70 [0.64–0.87]	0.949
Red blood cells, cell/μL	3.6 [2.6–5.1]	4.0 [3.6–4.3]	0.393
White blood cells, cell/μL	3.8 [3.4–4.2]	4.0 [3.3–5.4]	0.579
Platelets, cell/μL	78 [45–106]	91 [62–112]	0.136
Erythrocyte sedimentation rate, mm/h	12.5 [11–25]	11.5 [7.0–26.0]	1.000
Splenic length, cm	16.1 [14.2–19.9]	14.7 [13.3–16.5]	0.112

* Mann–Whitney test was used.

**Table 2 microorganisms-11-01017-t002:** Drugs used by patients with small intestinal bacterial overgrowth (SIBO) and without it.

Drug	Patients with SIBO (*n* = 26)	Patients without SIBO (*n* = 24)	*p*
Beta-blockers	19.2%	12.5%	0.704
Ornithine-aspartate	15.4%	16.7%	1.000
Diuretics	53.8%	37.5%	0.272
Ursodeoxycholic acid	15.4%	25.0%	0.490
Steroids	7.7%	25.0%	0.132
Antiviral drugs	30.7%	16.7%	0.327

**Table 3 microorganisms-11-01017-t003:** Assessment of risk factors for death in our patients.

Factor †	*p* ‡	Hazard Ratio
Small intestinal bacterial overgrowth	0.028	4.2 [1.2–14.9]
Esophageal varicese	0.104	
Ascites	0.316	
Hepatic encephalopathy	0.336	
Total bilirubin	0.551	
Albumen	0.027	0.84 [0.72–0.98]
Prothrombin (quick test)	0.540	

† The components of the Child–Turcotte–Pugh scale and esophageal varicese were selected as predictors. We have taken total bilirubin, prothrombin and albumin quantitatively, and ascites, hepatic encephalopathy and esophageal varicese veins by stages. ‡ A Cox regression model was used.

## Data Availability

Data available upon request due to restrictions.

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
