# Peer review of "Small Intestinal Bacterial Overgrowth Is Associated with Poor Prognosis in Cirrhosis"

_microorganisms, 2023, doi:10.3390/microorganisms11041017_

Round 1
Reviewer 1 Report
Current paper explored the role of small intestinal bacterial overgrowth in the prognosis of liver cirrhosis. This is an interesting idea. But several limitations should be noted from the current paper.
1. The ethical number should be provided.
2. “……There was no significant difference in the drugs used between these groups.” Which drugs had been performed to patients? Please provided a table.
3. The detail data should be provided in the Table 1. For example, Table 2 included Total bilirubin data but Table 2 did not include.
4. How many patients developed gastrointestinal bleeding?
5. The cause of death should be provided.
6. During the follow-up, whether patients received related drugs, which could improve small intestinal bacterial overgrowth?
7. The sample size is small, which could significantly influence the stable of these results.
Author Response
Current paper explored the role of small intestinal bacterial overgrowth in the prognosis of liver cirrhosis. This is an interesting idea. But several limitations should be noted from the current paper.
- The ethical number should be provided.
Authors response:
Dear Reviewer. Thank you very much for your comment. We added to the Methods section:
“The study was proved by institutional ethics board (protocol â„–03-16).”
- “……There was no significant difference in the drugs used between these groups.” Which drugs had been performed to patients? Please provided a table.
Authors response:
Dear Reviewer. Thank you very much for your comment. We added the new Table 2.
- The detail data should be provided in the Table 1. For example, Table 2 included Total bilirubin data but Table 2 did not include.
Authors response:
Dear Reviewer. Thank you very much for your comment. We added these date in Table 1.
- How many patients developed gastrointestinal bleeding?
Authors response:
Dear Reviewer. Thank you very much for your comment. We added to the Result section:
“The causes of death in all patients were acute-on-chronic liver failure, the development of which in 75% of cases was preceded by bleeding from esophageal varices.”
- The cause of death should be provided.
Authors response:
Dear Reviewer. Thank you very much for your comment. We added in the Result section:
“The causes of death in all patients were acute-on-chronic liver failure, the development of which in 75% of cases was preceded by bleeding from esophageal varices.”
- During the follow-up, whether patients received related drugs, which could improve small intestinal bacterial overgrowth?
Authors response:
Dear Reviewer. Thank you very much for your comment. We did not recommend our patients taking any anti-SIBO drugs, but we cannot guarantee that patients did not take them as prescribed for another reason. Since SIBO usually recurs in cirrhosis, it would most likely return several weeks or months after taking these drugs.
- The sample size is small, which could significantly influence the stable of these results.
Authors response:
Dear Reviewer. Thank you very much for your comment. Unfortunately we cannot fix this. However, this size did not prevent us from getting significant results. Since our work is the first to describe the effect of SIBO on long-term prognosis in cirrhosis, we hope that it will stimulate other researchers to conduct larger studies to verify our findings.
Reviewer 2 Report
Thanks for the chance to review this interesting study looking at the prognosis in a small cohort of cirrhotic patients with and without SIBO, showing reduced survival in those with SIBO. The study is well thought out and deserves publication. A few suggestions follow:
1. The methods section lacks sufficient detail. Please include more details about recruitment, primary and secondary outcomes, variables of interest, loss to followup and approach to missing data. Reference to the STROBE checklist for cohort studies (https://www.strobe-statement.org/download/strobe-checklist-case-control-studies-pdf) would be a recommended approach.
2. The inclusion criteria as written suggest all patients had biopsy proven cirrhosis. Is this correct?
3. Were any patients lost to followup?
4. "The exit of the mortality curve of patients with decompensated cirrhosis to a plateau 206 after 1 year is very interesting" line 206 - this phrase is unclear and needs to be expressed more clearly.
5. In the conclusion (line 229) the authors state that the prognosis is the first year in worse in those decompensated patients with SIBO and later prognosis is worse in those without decompensation. While there is some support for this conclusion in their data, as this is likely an ad hoc analysis of a subsequent hypothesis (ie it does not appear to be the primary objective or outcome studied, although no primary outcome is listed), the suggestion should be more tentative and requires further study.
Overall with an expanded methods section and more nuanced analysis I think the study should be published.
Author Response
Thanks for the chance to review this interesting study looking at the prognosis in a small cohort of cirrhotic patients with and without SIBO, showing reduced survival in those with SIBO. The study is well thought out and deserves publication. A few suggestions follow:
- The methods section lacks sufficient detail. Please include more details about recruitment, primary and secondary outcomes, variables of interest, loss to followup and approach to missing data. Reference to the STROBE checklist for cohort studies (https://www.strobe-statement.org/download/strobe-checklist-case-control-studies-pdf) would be a recommended approach.
Authors response:
Dear Reviewer. Thank you very much for your comment. We edited the manuscript in accordance with STROBE checklist and your recommendations.
- The inclusion criteria as written suggest all patients had biopsy proven cirrhosis. Is this correct?
Authors response:
Dear Reviewer. Thank you very much for your comment. Yes, is correct.
- Were any patients lost to followup?
Authors response:
Dear Reviewer. Thank you very much for your comment. We added in the Result section:
«All data on patient survival during the entire follow-up period were obtained.»
- "The exit of the mortality curve of patients with decompensated cirrhosis to a plateau 206 after 1 year is very interesting" line 206 - this phrase is unclear and needs to be expressed more clearly.
Authors response:
Dear Reviewer. Thank you very much for your comment. If you look at Figure 2c, you will see that patients with decompensated cirrhosis died rapidly during the first year of follow-up. Then they did not die during the second and third years of follow-up (the mortality curve stopped declining and flattened), after which they began to die again during the fourth year of observation, and the mortality curve went down again.
We have edited this sentence for better understanding:
« The transition of the mortality curve of patients with decompensated cirrhosis to a plateau after 1 year is very interesting. The presence of SIBO determines mortality of these patients during the first year of follow-up, but then this ceases to have a significant impact on their prognosis. It is possible that a state of intestinal permeability or the absence of significant gut dysbiosis protects those patients with decompensated cirrhosis and SIBO, who survived the first year of follow-up, from the harmful effects of SIBO.»
- In the conclusion (line 229) the authors state that the prognosis is the first year in worse in those decompensated patients with SIBO and later prognosis is worse in those without decompensation. While there is some support for this conclusion in their data, as this is likely an ad hoc analysis of a subsequent hypothesis (ie it does not appear to be the primary objective or outcome studied, although no primary outcome is listed), the suggestion should be more tentative and requires further study.
Authors response:
Dear Reviewer. Thank you very much for your comment. We added in the Conclusions section:
“According to our preliminary data, SIBO affects the prognosis only in the first year of follow-up in decompensated cirrhosis, and only in subsequent years in compensated cirrhosis. Further studies are required to verify our results in a larger cohort of patients and to select the optimal drugs and regimes for their therapeutic and prophylactic use in order to improve the prognosis of cirrhosis patients with SIBO.”
Round 2
Reviewer 1 Report
The most comments had been resolved. However, the remain 2 comments should be further considered.
1. The number of patients with gastrointestinal bleeding should be added to the table 1.
2. If the data regarding the patients received related drugs, which might influence the SIBO, during the follow-up was unclear, author should state it in the limitation section.
No other comments.
Author Response
The most comments had been resolved. However, the remain 2 comments should be further considered.
1. The number of patients with gastrointestinal bleeding should be added to the table 1.
Authors response.
Dear Reviewer. Thank you for your suggestion. These data are added to the Result section.
"Fatal bleeding from esophageal varices occurred in 9 (35.6%) patients with SIBO and in 3 (12.5%) patients without SIBO (p=0.066)."
We did not include these data in Table 1, as it describes the data at the time of inclusion in the study.
2. If the data regarding the patients received related drugs, which might influence the SIBO, during the follow-up was unclear, author should state it in the limitation section.
Authors response.
Dear Reviewer. Thank you for your suggestion. These data are added to the limitation subsection.
"The forth limitation is that we could not guarantee that patients during the follow-up period could take drugs that eliminate SIBO. However, given that SIBO tends to recur, we hope that this limitation did not have a significant impact on our study results."
Reviewer 2 Report
I am happy with the revised paper.
Author Response
Authors' response.
Dear Reviewer. Thank you for appreciating our work.